# Investigation of Chemical Compositions and Biological Activities of *Mentha suaveolens* L. from Saudi Arabia

**DOI:** 10.3390/molecules27092949

**Published:** 2022-05-05

**Authors:** Bashayr Aldogman, Hallouma Bilel, Shaima Mohamed Nabil Moustafa, Khaled F. Elmassary, Hazim M. Ali, Faddaa Qayid Alotaibi, Mohamed Hamza, Mohamed A. Abdelgawad, Ahmed H. El-Ghorab

**Affiliations:** 1Department of Chemistry, College of Science, Jouf University, Sakaka 72341, Saudi Arabia; 401205881@ju.edu.sa (B.A.); hbilel@ju.edu.sa (H.B.); hmali@ju.edu.sa (H.M.A.); faddaa879@gmail.com (F.Q.A.); 2Department of Biology, College of Science, Jouf University, Sakaka 72341, Saudi Arabia; shymaa.nabil@ju.edu.sa (S.M.N.M.); mhabdelhameed@ju.edu.sa (M.H.); 3Flavour and Aroma Department, National Research Centre, Giza P.O. Box 12622, Egypt; kfarouk@yahoo.com; 4Department of Pharmaceutical Chemistry, College of Pharmacy, Jouf University, Sakaka 72341, Saudi Arabia; mhmdgwd@ju.edu.sa

**Keywords:** *Mentha suaveolens* L., carvone, rosmarinic acid, antioxidant, antifungal activity

## Abstract

*Mentha* is an aromatic plant used since antiquity for its pharmaceutical virtues. The climate of Saudi Arabia favors the growth of aromatic plants including *Mentha suaveolens* L. The aim of this study is to analyze the volatile oils of different parts of fresh and dried *Mentha suaveolens* L. grown in Saudi Arabia (Aljouf area) using Gas Chromatography/Mass Spectrometry (GC/MS) and Gas Chromatography Flame Ionization Detector (GC/FID) techniques, to recognize the effect of drying on chemical composition, then to evaluate the antioxidant and antifungal activities of different extracts. In total, 118 compounds were identified via GC/MS and GC/FID, in which carvone is the main volatile constituent (stems, leaves, whole plant 45–64%). This investigation deduces that *Mentha* belonged to the carvone chemotype. Then, the analysis of non-volatile constituents of fresh and dried *Mentha* was performed by HPLC. The main phenolic compound of fresh and dried *Mentha* for different parts was rosmarinic acid (ranging from 28,002.5 to 6558 µg/g). The ethanolic extract of fresh stem showed the highest antifungal activity (53% inhibition) compared with miconazole (60% inhibition) but the ethanoic extract of dry stem showed no activity. Additionally, all ethanolic extracts, whether for fresh or dry *Mentha*, have antioxidant activity more than 90% while the antioxidant activity of whole plant volatile oil is equal to 53.33%. This research shows that *M. suaveolens* L. could be applied to manufacture natural antioxidants, antifungal, and flavoring agents.

## 1. Introduction

*M. suaveolens L*. is common plant in Saudi Arabia known for its strong antifungal, anti-aflatoxigenic, and antioxidant potential [1]. *Mentha* species have been well recognized for their aromatic and medicinal purposes since ancient times [2]. Recently, several *Mentha* species gained increasing interest for their potential use as natural food preservatives because of their strong antifungal, anti-aflatoxigenic, and antioxidant properties [1,3]. These biochemical properties are mainly due to the presence of several aromatic and phenolic compounds in different parts of the *Mentha* species. The chemical composition of Mentha species differs according to the growing season [4,5]. Additionally, the volatile oil composition of *Mentha* species from various populations and geographical regions indicated that the plants relate to either carvone or menthol or linalool chemotypes [6]. Several *Mentha* species were investigated in the Kingdom of Saudi Arabia. Burham et al. [7] analyzed the chemical composition of the volatile oils extracted from the leaves of *M. longifolia* harvested from the Albaha area in Saudi Arabia using GC/MS. As well as the compound categories identified, the content of oxygenated monoterpenes was the highest in comparison with other classes. A total of 46 compounds were detected, the main oxygenated monoterpene component was piperitone (30.77%). Caryophellene was the major sesquiterpene hydrocarbon identified (5.58%), while γ-terpinene was the main monoterpene hydrocarbon (1.36%) [7]. Abdel-Hameed et al. [8] examined the effect of extraction method on its yield and chemical composition for *M. longifolia* harvested from Taif, Saudi Arabia. The main chemical components identified were monoterpenes and sesquiterpenes; pulegone was the most prevalent [8]. The drying of aromatic plants is an essential step of the manufacture process to ensure the high quality of the end products [9]. Two *Mentha* varieties cultivated in Saudi Arabia, Medina, and Hasawi, were found to be high in volatile oils and phenolic content, especially in the soluble fractions. Regarding outside KSA, previous studies have shown that the major components of Moroccan *M. suaveolens* essential oils were piperitenone oxide (56.28%), piperitenone (11.64%), and pulegone (6.16%) [2]. Additionally, the study of the chemical composition in various regions of Morocco showed that the chemical composition changed and the main compounds were different: piperitenone oxide (56.00%), P-cymenol-8 (20.60%), caryophyllene (5.70%); piperitone oxide (26.00%), piperitenone oxide (25.00%), caryophyllene (9.80%); pulegone (50.00%), P-cymenol-8 (10.40%), and borneol (5.60%) [10]. The study of the Italian *M. suaveolens* showed that the piperitenone oxide was the main component [2]. Two different populations of *M. suaveolens* from Eastern Iberian Peninsula were studied and the results showed that the major components are different and correspond to two different chemotypes: piperitenone oxide (35.2–74.3%) and piperitone oxide (83.9–91.3%) [11]. A study of the chemical composition of Egyptian *M. suaveolens* showed that the major component was carvone (50.59%) followed by limonene (31.25%) [5]. Based on the results of these studies, this difference in chemical composition between the different geographic localities can be explained by many factors such as fertilization, drying effect, climatic change, and others.

*M. suaveolens* is an aromatic herb native to Southern and Western Europe. *M. suaveolens* is a perennial, herbaceous plant characterized by a sweet scent. The plant can grow up to 100 cm in height [12]. *M. suaveolens* cultivated in Egypt is rich in oxygenated compounds, with carvone and limonene being the predominant compounds, followed by hydrocarbons [4,5]. El-Kashoury et al. [4] identified two new triterpenes (3*β*-acetyl-22α-hydroxy ursa-12,20-diene and 2*α*, 3*β*-dihydroxy-olean-18-en-29-oic acid) in the ethanolic extracts of the aerial parts of *M. suaveolens* growing in Egypt. Moreover, seven known compounds were identified: *β*-sitosterol, *β*-sitosterol-3-*O*-β-D-glucoside, oleanolic acid, dihydrolimonene, 7-hydroxy-*p*-cymene, isoquercitrin, and rutin [4]. In Morocco, GC/MS analysis of the volatile constituents extracted from *M. suaveolens* L. revealed that the major components were piperitenone, pulegone, and piperitone [13]. In another study, the chemical composition of the volatile constituents extracted from 10 wild populations of *M. suaveolens subsp*. *timija* was analyzed using GC/MS analysis. Collectively, 44 compounds were identified in all samples with a percentage that reached at least 97.3% of the oil chemical composition. The major components identified were menthone, pulegone, cis-piperitone epoxide, piperitone, trans-piperitone epoxide, piperitenone, piperitenone oxide, (*E*)-caryophyllene, germacrene D, isomenthone, and borneol [14]. In a recent study, Benali et al. [15] investigated the chemical composition of the volatile constituents extracted from *M. suaveolens* in Morocco using GC/MS analysis. The main compounds identified were piperitenone oxide, germacrene D, *β-trans*-caryophyllene, piperitone oxide, 1-4-terpineol, and δ-terpinene, among a total of 17 compounds identified [15]. The major phenolic compounds identified by HPLC in different Mentha varieties are caffeoylquinic acid, salvianic acid, rosmarinic acid, luteolin, salvigenin, chrysoeriol, thymonin, carnosol [6]. Elansary et al. [16] investigated phenolic compounds of M. piperita and stated that rosmarinic acid, cryptochlorogenic acid, and chlorogenic acid are the major compounds. The non-volatile extract of *Mentha* plant extracts are rich sources of phenolic compounds and flavonoids [16]. The most important and frequently encountered phenolic acids in *Mentha* species are caffeic acid and its derivative compounds and other acids such as cinnamic acid, gentisic acid, protocatechuic acid, hydroxybenzoic acid, and vanillic acid [17]. Antioxidants are of major interest to scientists, however, many carcinogenic and toxic effects were reported for synthetic antioxidants, making natural antioxidants promising alternatives [18]. The volatile oils of *M. suaveolens* have several proven and beneficial biological activities, especially antioxidant activity [4]. Other studies also showed that the volatile oils extracted from *M. suaveolens* exhibited potent antifungal activity against *Aspergillus niger* and *Candida albicans* [5].

To our knowledge, there is no report on the chemical composition and biological activities of the volatile oils of *M. suaveolens* L. native to Saudi Arabia (Aljouf area). Therefore, the objective of this study was to extract non-volatile and volatile oils from *M. suaveolens* L. and to identify the chemical constituents using GC, GC-MS and high-performance liquid chromatography (HPLC). Moreover, biological activities (antioxidant and antifungal activities) are presented in this study.

## 2. Results

### 2.1. Analysis of Volatile Oils of Fresh and Dried M. suaveolens L. by GC/MS and GC/FID

The volatile constituents obtained from *M. suaveolens* L. under study were pale yellow with a pleasant and distinct odor. The total yield percentages of volatile constituents from *M. suaveolens* L. of fresh and dried whole plant extracts were 0.57 ± 0.03% and 0.225 ± 0.01%, respectively. The total yield percentages of fresh and dried leaf extracts were 0.235 ± 0.012% and 0.24 ± 0.016%, respectively. The total yields of fresh and dried stem extracts were 0.24 ± 0.016% and 0.9 ± 0.05%, respectively.

The analysis of volatile oils of fresh Mentha was performed by GC/MS and GC/FID; 125 volatile constituents of fresh Mentha were identified. The percentages of chemical identification of whole plant, leaf, and stem extracts were 99.14%, 97.35%, and 96.58%, respectively. The main volatile constituent of Mentha in whole plant, leaf, and stem extracts was carvone at 43.65%, 64.31%, and 58.8%, respectively. The other major volatile constituents in whole plant, leaf, and stem extracts were myrcenol (5.88%, 3.93%, and 3.85%, respectively), terpineol<1-> (4.29%, 5.61%, and 4.4%, respectively), pulegone (3.8%, 2.1%, and 2.2%, respectively), and limonen-10-ol (2.79%, 1.24%, and 0.15%, respectively). All components are listed in Table 1.

The analysis of dried Mentha volatile oils was completed by GC/MS and GC/FID. In total, 125 volatile constituents of dried Mentha were identified. The percentages of these chemical compounds of whole plant, leaves, and stems were 99.98%, 100%, and 99.28%, respectively. The major compound found in whole plant, leaf, and stem extracts was carvone at 45%, 53.45%, and 49% respectively. The other major volatile constituents in whole plant, leaves, and stems were copaene (β) (11%, 3.81%, and 0.08%, respectively), aromadendrene (allo) (3.63%, 0.12%, and 0.05%, respectively), octen-2-ol (3.22%, 0.18%, and 0.08%, respectively), epi-α-muurolol (3.07%, 2.67%, and 2.44%, respectively). All constituents are reported in Table 1.

### 2.2. Drying Effect on Volatile Oil Constituents of Mentha

A variation in chemical composition concentration for some compounds was found due to the drying effect as shown in Figure 1. As shown in the figure, the LOC is the highest concentration in all plant extracts. While the S and the HOC concentration increases after the drying process in all plant extracts, no significant changes were found in the concentration of the M after the drying process.

### 2.3. Chemical Composition of Non-Volatile Constituents of Fresh and Dried Mentha by Using HPLC

The analysis of fresh *Mentha* non-volatile constituents was performed by HPLC. The major phenolic derivatives identified in the ethanolic extract from the fresh whole plant were rosmarinic acid (2223.3µg/g), rutin (676.7 µg/g), and ferulic acid (226.7 µg/g). Additionally, the main phenolic derivatives found in fresh leaves of *Mentha* were rosmarinic acid (28,002.5 µg/g), rutin (3383.8 µg/g), and ferulic acid (1520 µg/g). In fresh stems of *Mentha*, the main phenolic derivatives identified were rosmarinic acid (6558µg/g), catechin (1340.4 µg/g), and naringenin (371.8 µg/g). The other components are listed in Table 2. In addition, the analysis of dried *Mentha* non-volatile constituents was performed by HPLC. The major phenolic derivatives identified in the ethanolic extract from the dried whole plant of *Mentha* were rosmarinic acid (21,191.9 µg/g), rutin (252.16 µg/g), and quercetin (153.8 µg/g). The main phenolic derivatives found in dried leaves of *Mentha* were rosmarinic acid (15,165.1 µg/g), rutin (194.7 µg/g), and quercetin (156.98 µg/g). Moreover, the main phenolic derivatives found in dried stems of *Mentha* were rosmarinic acid (8378.4µg/g), naringenin (130.3 µg/g), and catechin (68.1 µg/g). The other components are recorded in Table 2.

### 2.4. Biological Activities of Mentha suaveolens

#### 2.4.1. Antioxidant Activity of Volatile Oils and Non-Volatile Extracts of Fresh and Dried *Mentha suaveolens* L.

Antioxidant activities of *Mentha* were evaluated using 1,1-diphenyl-2-picrylhydrazyl (DPPH) method. DPPH radical scavenging activity of the positive standard Butylated hydroxytoluene (BHT) and plant extracts were expressed by inhibition percentage (%). The ability of the tested extract to act as an electron donor in the conversion of DPPH• into DPPH-H was studied.

##### Antioxidant Activity of Volatile Oils from *Mentha suaveolens* L.

The percentages of inhibition were measured for all the samples presented in Figure 2. The results showed that the antioxidant activity of samples ranged from 31% to 53%.

##### Antioxidant Activity Non-Volatile Constituents from *Mentha*

DPPH• (purple-colored) was reduced to DPPH-H (yellow colored) by using tested samples. The obtained results showed that all the ethanolic samples have good antioxidant activity with values exceeding 90%. The inhibition% measured for all samples is presented in Figure 3.

#### 2.4.2. Antifungal Activity of Fresh and Dried *M. suaveolens* L.

Isolated fungi of the studied white cheese samples show that a total of 13 fungal isolates were recovered on PDA media. The isolated fungi were identified as seven species belonging to four genera. The sequence of the most dominant species was closely related to *Penicillium glabrum* (Genbank accession number, MW534476.1) with 100% similarity https://www.ncbi.nlm.nih.gov/nuccore/MW534476.1 (2 February 2021).

*Penicillium* sp. was the most predominant genus encountered in 37.4% of the total fungi recovered on PDA media. Two species of Penicillia were most common, *P. glabrum* was the most prevalent (22.4% of the total fungi) and *P. aurantiogriseum* (14.9%). Three species of Aspergilla were identified, *A. niger* was the most prevalent (11.21% of the total fungi), followed by *A. ustus* (9.3%), and *A. fumigatus* (7.47%). *Geotrichum* was isolated from one sample only cultivated on PDA medium.

##### Antifungal Activity of Volatile Oils from *M. suaveolens* L.

Antifungal activity was estimated by determining the capacity of inhibition of mycelial growth of the fungi species under study. Leaves of fresh *Mentha* presented the highest biological activity compared with the other extracts from *M**entha,* as shown in Figure 4. In total, 20 mg\ml of fresh and dried *Mentha* leaf extracts prevented the growth of the fungi by 51.18 ± 0.15 and 47.06 ± 0%, respectively. Additionally, the fungi can be inhibited by using fresh and dried stems of *Mentha* extracts, and the percentage is 48.53 ± 0.15% and 45.44 ± 0.23%, respectively, followed by inhibition by the whole plant, fresh and dried *Mentha* extracts, with the percentages of 41.53 ± 0.23% and 35.29 ± 0%, respectively.

##### Antifungal Activity of Non-Volatile Constituents from *Mentha*

Antifungal activity was estimated by determining the capacity of inhibition of mycelial growth of the species under study. Fresh stem extracts of *Mentha* had the highest biological activity compared with the other extracts from *Mentha*. A concentration of 20 mg/mL of fresh stems, leaves, and whole plant extracts of *Mentha* can prevent the growth of the fungi by 52.94 ± 0.24%, 41.47 ± 0.05%, and 39.12 ± 0.096%, respectively, as shown in Figure 5. The other extracts, including dried (whole plant, leaves, and stems), of *Mentha* did not affect the growth and sporulation of the tested fungi.

Antifungal activity of volatile oils and non-volatile constituents of *Mentha*, examined using analysis of variance and presented in Table 3, shows that mean squares were highly significant for treatments.

## 3. Discussion

The volatile oils obtained from *Mentha* under study were pale yellow with a pleasant and distinct odor. The total yield percentages of volatile oils from different parts of fresh and dried *Mentha* varied from 0.225% to 0.9%. Our results were in close agreement with Petretto et al. [29] and Kasrati et al. [14], who reported that the volatile oil yields of the aerial parts of *M. suaveolens ssp. insularis* from Sardinia was 0.2%. The non-volatile constituents obtained from *Mentha* under study were yellow to deep green with a pleasant and distinct odor. The total yield percentages of non-volatile constituents from different parts of fresh and dried *M**entha* varied from 0.6% to 3.4%. Our results agree with Barchan et al. [30] who found that the content of non-volatile constituents of *M. pulegium* and *M. piperita* from Morocco to be 3.4% and 4%, respectively. In contrast, non-volatile constituents were found in high levels (5%) in the extracts of three Algerian mints [31]. Moreover, El-Ghorab [32] reported that non-volatile constituents of Egyptian Mentha were 4.5%.

The volatile constituents of fresh and dried *Mentha* from Sakaka belonged to the carvone chemotype. Our results agree with El-Kashoury et al. [4], who reported that the carvone was the main compound in *Mentha* in different seasons during the year (56% to 31%). Other studies are in agreement with our results where *Mentha* was classified by the carvone pathway [6,33]. Hussain et al. [34] and Boukhebti et al. [35] analyzed the chemical composition of *Mentha* from Pakistan and Algeria, in which the main compound in both studies was carvone (59.5% and 59.4%, respectively). Although the carvone percentage in leaves was higher (64.31%) than that reported by Hussain et al. [34], the percentage was lower in whole plant (43.65%) compared to Boukhebti et al. [35]. These differences in carvone concentration might be due to different environmental conditions. In another study of fresh *Mentha* in Morocco, pulegone was the major compound, accounting for 17.61% [13], while in our study, the percentage was 3.8% in the whole plant, 2.1% in leaves, and 2.2% in stems. Additionally, Sutour et al. [36] reported a higher level of pulegone in their samples (44.4%) compared to our results. In another study of dried *Mentha* in Morocco (Iguer region), pulegone was absent, agreeing with our results [14]. These changes in the concentration of pulegone might be due to the different habitats from which plants were collected. In our results, α-Terpineol concentration in fresh *M**entha* whole plant (0.15%) was lower than that reported by Salhi et al. (0.4%) [37]. In another study of *M. suaveolens ssp. insularis* in Sardinia, the percentage of the linalool was 1.37%, slightly lower than what was observed in this study for fresh whole plants (1.54%) [29]. In an recent study, α-Humulene was found in a high concentration (1.09%) in dried *Mentha* (Morocco) whole-plant extract compared to our results (0.04% in dried whole plant) [15]. In the same study, terpinene (γ) was among the major compounds accounting for 5.33% in dried whole-plant extract, which is a much higher percentage than the one we observed (0.03% in dried whole-plant extract).

The chemical compounds in fresh and dried *Mentha* analyzed by GC/MS can be categorized into four chemical classes: M, LOC, HOC, and S as shown in Table 2. The chemical classes were classified in the following decreasing order: high percentage of LOC due to its high content of carvone in all plant extracts, followed by S, HOC, and M in all different extracts of fresh and dried *Mentha*.

The above results (Table 2) were similar to those reported by Petretto et al. [29] during their work on fresh *M. suaveolens ssp. insularis* in Sardinia, especially high percentages were reported for oxygenated compounds (87.1%) in the whole plant extract. Moreover, the percentage of sesquiterpenes (14.2%) was similar to the percentage reported in the fresh whole plant extract (15.83%) in our study [29]. El-Kashoury et al. [4] analyzed the chemical composition of fresh *M. suaveolens Ehrh.* in Egypt. The highest percentage of oxygenated compounds was 62.9% in their whole plant extract, which was much lower than the percentages reported in our study. Our results were in close agreement with Hussain et al. [34], who identified oxygenated compounds in dried leaves of Mentha from Pakistan occurring at 81.5%. In the same study, sesquiterpenes were found at 6.1% in dried leaf extract compared to 12.90% in our analysis. However, the monoterpenes percentage was (1.15%) lower than reported in their study (9.09%) [34]. In another study of dried Mentha in Taif from KSA, the percentage of the oxygenated compounds was 91.9%, which was slightly higher than our results [8]. In contrast, Burham et al. [7] reported a much lower percentage of oxygenated compounds in dried leaves (30.4%) of Mentha from Albaha Area southern KSA. They also reported higher percentages of monoterpenes and sesquiterpenes (54.3% and 26.08%, respectively) compared to our observations in leaf extracts (1.15% and 12.90%, respectively) [7].

Based on the above results, significant change in the chemical composition of volatile oils might be due to different plant parts, environmental factors, seasons, geographical location, and plant age of the *Mentha* plant.

Previous investigation of the chemical composition of the volatile oils from *Mentha* in Senegal showed that the pulegone rate dropped in the oils of the dried plants [38]. Similarly, our results showed a sharp drop in pulegone concentration to very low or zero levels in dried *M. suaveolens* L. Kohari et al. [9] reported the drop in α-Terpineol percentage in the dried Japanese *peppermint* (0.27% in fresh to 0.18% in dried). This result was in close agreement with our results. The 1,8-cineole concentrations in *Mentha* from Senegal were lower in the dried plants, which was similar to our observations [38]. Moreover, in our results, the percentages of monoterpenes after drying the whole plant and leaves decreased. Similar results were previously reported [9,39]. In a study of *Mentha* from Senegal, monoterpene levels increased from 3.4% to 4.7% after drying [38]. Sesquiterpenes were also increased after drying *M. suaveolens* L. in our analysis, which also agrees with the studies of Diop et al. [38] and Ahmed et al. [39]. Different factors could contribute to the observed variation in volatile constituents. One of these factors is the chemical properties of volatile oils, mainly their structure and volatility. The type of plant is also a significant factor in this regard. There is also a chance for new compounds to form based on the chemical reactions involved, such as glycoside hydrolysis, oxidation, esterification, etc. [39]. Changes in volatile compound concentrations after drying have been reported and explained by the continuous biosynthesis after harvest [40].

Egyptian and Moroccan essential oils of *M. suaveolens* L. were studied and the results show that β-copaene is present in low amounts (less than 1%) [4,41]. Additionally, β-copaene had been detected in some Algerian species of Mentha with a concentration ranging from 0.82 to 0.9% [42]. Boukhebti et al. showed that the content of β-copaene in the essential oil of Mentha spicita was 0.347% [35]. Another study conducted in the USA showed that β-copaene was present in Peppermint EO (0.07%), Native Spearmint (0.25%), and Scotch Spearmint (0.18%) [43]. In this study, α and β copaene were detected in whole plant, leaves, and stems where the concentration of α copaene was 0.57%, 0.8%, and 1.37%, respectively and β copaene was 11%, 3.81%, and 0.08%, respectively.

Non-volatile components of fresh and dried Mentha were in close agreement with those reported by Elansary et al. [16] and Mišan et al. [44]. They investigated the phenolic compounds of *Mentha* species and reported, rosmarinic acid as a main compound. In comparison, Kulig et al. [45] reported that rutin was the primary phenolic acid in *Mentha* from the Slovak Republic, which was the case in our study for the fresh and dried whole plant and fresh and dried leaf extracts. Additionally, Farnad et al. [17] analyzed *Mentha* from West Azerbaijan and found that the main compounds were chlorogenic acid and rutin. Previous studies on *Mentha* species revealed some phenolic derivatives in the genus, such as rosmarinic acid, caffeic acid, ferulic acid, and catechin [31,43,44]. The highest antioxidant activity (53.33%) was observed in the whole-plant extract of dried *Mentha*, which can be explained by its high carvone content (45%), and which is known for its antioxidant activity. Indeed, Elmastaş et al. [46] reported that the antioxidant activity of carvone is equal to 95%. Additionally, it contains a high concentration of copaene (β) (11%), which has high antioxidant activity [47]. The mild antioxidant activity of the volatile oils of *Mentha* may be due to its weak phenolic compounds. Approximate results were obtained by Bardaweel et al. [48], who showed that volatile oils have weak antioxidant activity.

Good antioxidant activity of non-volatile constituents of *Mentha* was observed (94.23 ± 0.007% for the fresh whole plant, 94.50 ± 0.005% for fresh leaves, 94.27 ± 0.01% for fresh stems, 92.83 ± 0.02% for the whole dried plant, 90 ± 0.014% for dried leaves, and 93.33 ± 0.007% for dried stems). These results agree with Mata et al. [49], who reported that ethanol extracts have good antioxidant activity.

The high antioxidant activity of fresh whole plant can be explained by its high content of rosmarinic acid (2223.3 µg/g) [50,51], rutin (676.7 µg/g) [52], and ferulic acid (226.7 µg/g) [53], which are known for their high antioxidant activity. The large effect of fresh leaves was attributed to the high content of rosmarinic acid (28,002.5 µg/g) [50,51], rutin (3383.8 µg/g) [52], ferulic acid (1520 µg/g) [53], and caffeic acid (1141.5 µg/g) [53]. The phenolic compounds of fresh stems contain rosmarinic acid (6558 µg/g) [50,51], naringenin (371.8 µg/g) [54], and rutin (194.6 µg/g) [52], which also exhibited high antioxidant activity.

The good antioxidant activity for dried whole plant may be due to its high content of rosmarinic acid (21,191.9 µg/g) [50,51], rutin (252.16 µg/g) [52], and naringenin (53.22 µg/g) [54]. The high antioxidant activity of dried leaves was due to the high content of rosmarinic acid (15,165.1 µg/g) [50,51] and rutin (194.7 µg/g) [52]. The phenolic compounds of dried stems contain rosmarinic acid (8378.4 µg/g) [50,51], naringenin (130.3 µg/g) [54], and rutin (41.6 µg/g) [52], which have high antioxidant activity.

The high antifungal activity of volatile oils of *Mentha* can be explained by its high content of carvone, which showed high antifungal activity [55]. The volatile oils of fresh *Mentha* contain pulegone, which also exhibited high antifungal activity [56]. Additionally, it contains 1,8-cineole, which showed good antifungal activity but was lower than carvone [57].

The high antifungal activity was observed for the non-volatile constituents of fresh extracts of *Mentha* due to the high concentration of rosmarinic acid, catechin, and luteolin, known for their high antifungal activity [58,59,60]. Moreover, the fresh stem extracts of *Mentha* contained a high concentration of eugenol, which showed high antifungal activity [61].

## 4. Materials and Methods

### 4.1. Chemicals and Plant Material

All standard (authentic compounds of essential oils (Decan, 1,8 Cineole, Ocimene (Ε-β), Benzyl formate, Terpinolene, Linalool, Myrceno, Terpineol<1->, Terpinen-4-ol, α-Terpineol, γ-Terpineol, Carveol (Trans Carveol(cis), Carvone, Pulegone, Elemene (δ-), Copaene (α), Aromadendrene) and phenolic compounds (Protocatechuic acid, p-hydroxybenzoic acid, Catechin, Vanilic acid, Cinnamic acid, Naringenin, Eugenol, Caffeic Acid, Coumaric acid, Ferulic acid, Rutin, Luteolin, Quercetin and Rosmarinic acid)) and chemical reagents were supplied from Sigma Aldrich company (Burlington, MA, US and used without further purifications.

Fresh *M**. suaveolens* L. plants were collected from a local farm in Sakaka, Aljouf, Saudi Arabia, in September 2020. Plant identification was performed by Hamdan A., Al-Jouf, KSA.) the herbarium (59-CPJU) voucher specimen was stored at College of Pharmacy, Jouf University (Sakaka, Saudi Arabia). Fresh *M. suaveolens* L. plants were dried under shade at room temperature in a fume hood.

### 4.2. Extraction Methods

#### 4.2.1. Extraction of Volatile Oil of *M. suaveolens* L.

The volatile oils of different samples were extracted, 200 g of each sample, by hydro distillation using Clevenger apparatus (Clevenger Apparatus was supplied from Shiva Scientific Glass Private Limited, New Delhi, India.) during 5 h. Then, the volatile oils were extracted by dichloromethane (DCM) (3 × 50 mL) and dried by adding magnesium sulfate anhydrous. The DCM was filtered and removed in a rotary evaporator (Model: R-3001, Evaporating Flask: 500 mL, 1000 mL, Rotation Speed: 10–280 rpm, Temperature Range: Room temp +5 °C~95 °C. Evaporating Speed: 23.5 mL/min (Water) supplied from GWSI manufacture, Zhengzhou, China) and then stored at −4 °C in opaque containers [8].

#### 4.2.2. Extraction of Non-Volatile Constituents of *M. suaveolens* L.

Each sample of *Mentha* (100 g) was extracted with 99.8% ethanol (200 mL for each part), for 72 h, in a dark place and with stirring. Then, the obtained extract was filtered and dried by adding anhydrous magnesium sulfate. The rotary evaporator was used for solvent removal from extract and then stored in dark containers at −4 °C [32].

### 4.3. Analysis of Volatile Oils of M. suaveolens L. by GC and GC/MS

#### 4.3.1. GC Analysis

Volatile oils were analyzed using GC/MS and GC, instrumental details and condition according to the reported method [19]. See Appendix A.

#### 4.3.2. GC/MS Analysis

The quantitative determination and chemical composition of volatile oils were estimated using the adjusted reported method [14,15,16,17,18,19,20,21,22,23,24,62] (see Appendix A).

### 4.4. HPLC Analysis of Non-Volatile Compounds

The non-volatile extracts were analyzed using HPLC, instrumental details and condition according to the reported method [63]. All phenolic standards were prepared in methanol within the range from 0.5 to 100 µg/mL and these standards were used for quantitative and qualitative identification of phenolic compounds in *M. suaveolens* L. extract chemical composition depending on the standard calibration curve for each standard and retention time, respectively (see Appendix A).

### 4.5. Antioxidant Activity

The scavenging DPPH free radical ability was determined in vitro according to the procedure of Yue and Xu [64]. DPPH solution (0.1 mM, 1.8 mL) was mixed with 0.2 mL of each diluted Mentha extract stock solution (5 mg/mL) and incubated in the dark for 30 min at 25 °C. In this assay, the percentage of DPPH reduction by samples was compared to BHT. The experiment was repeated three times. The following formula was used to quantify radical scavenging activity:Inhibition (Scavenging effect) (%) = [A_0_ – A_1_)/A_0_] × 100(1)

A_0_ is the absorbance of the control reaction (t = 0 min).

A_1_ is the absorbance of the tested extract solution (t = 30 min).

### 4.6. Antifungal Activity

Ten samples of locally fabricated cheese were gathered arbitrarily from farmer’s houses of Al-Jouf–Sakaka in KSA. Samples were collected in (sterile, clean, and dry) containers and shipped to test centre within 1–2 h of collection (at 4 + 2 °C).

#### 4.6.1. Detection and Isolation of Fungi

The dilution-plate method [22] was used to isolate and identify fungi from the collected cheese samples. Samples were inoculated on potato dextrose agar (PDA) obtained from Oxoid [24]. Rose Bengal (30 ppm) was used as an antibacterial agent.

One gram of white cheese was suspended in 90 mL sterilized distilled H_2_O using a rotating shaker to homogenate the suspension. Then, 10 serial dilutions were prepared, and 1 mL of each dilution was inoculated into a petri dish. Then, melted medium was poured, mixed well, and left to solidify. After solidification, Petri dishes were incubated at 27 ± 2 °C for 5–7 days. Colonies were counted and isolated for purification and identification.

#### 4.6.2. Identification of the Isolated Fungi

Morphological identification of the isolated fungi was carried out based on their macro and microscopic characteristics using the taxonomic methods [28,63]. Additionally, the most dominant species were identified using molecular analysis of the ITS1-5.8S rRNA–ITS2 region (animal health research institute, Dokki, Giza, Egypt).

#### 4.6.3. Effect of Volatile Oils and Non-Volatile Extract on Mycelium Growth of Isolated Fungi

In total, 15 mL of PDA medium were placed in each Petri dish (9 cm diam.), and after solidification, a circular hole (2 cm, diam.) was formed. Then, 1 mL of each different concentration of the tested extract was added to each well. Each Petri dish was inoculated with four fungal species and incubated at 28 °C for seven days. PDA medium without additives was used as a negative control, whereas plates containing 20 mg/mL Miconazole Nitrate were used as a positive control. Each treatment was repeated three times and subjected to statistical analysis. The diameter of inhibition zones was measured for each well, and the results were statistically analyzed. In total, 1 mL of plant extract (20 mg/mL) was added to each well to determine the inhibition zone and inhibition percentage (%).

The diameter of inhibition zones was measured, and the percentage of antifungal activity was calculated according to Equation (2):(2)$$\mathrm{Inhibition}\;(\%)=\frac{\mathrm{DC}-\mathrm{DT}}{\mathrm{DC}}\times 100$$

DC = the diameter of fungal mycelium growth in the control Petri dish.

DT = the diameter of fungal mycelium growth in the treated Petri dish, containing Mentha extracts.

### 4.7. Statistical Analysis

Separated completely randomized designs as one-way ANOVA were conducted for statistical analysis procedure of the obtained data. Both antioxidant and antifungal activities of volatile oils and non-volatile extracts of *Mentha* were carried out in triplicate experiments. Each experiment included seven treatments (whole plant of fresh Mentha, whole plant of dried Mentha, leaves of fresh Mentha, leaves of dried Mentha, stems of fresh Mentha and stems of dried Mentha as well as the control treatment with BHT). Duncan’s multiple range test [65] was used to estimate the performance of mean treatments depending on significance of mean of squares according to the ANOVA table at a significance level of *p* = 0.05, where a different superscript letter refers to a significant difference among those treatments. MSTAT- Cv.2.10 software program package processed all data [66].

## 5. Conclusions

*M. suaveolens* L. from the Sakaka region was investigated for its chemical components and biological activities for the first time. Regarding the main chemical components of volatile oils, carvone dominated in all samples. The volatile oil *M. suaveolens* L. was carvne chemo type. Additionally, the main phenolic compounds in ethanolic extract were rosmarinic acid, rutin, catechin, and naringenin which are responsible for the antioxidant and antifungal activities of *Mentha*. It was found that copaene had the highest concentration in *M. suaveolens* L. in comparison with the same species in another locality. Fungi (*Penicillium glabrum*) isolated from white cheese was inhibited by *Mentha* extracts for the first time as well as fungi (*Penicillium glabrum*) isolated from white cheese that was inhibited by *Mentha* extracts for the first time.

From the previous results, the *M. suaveolens* L. has high antioxidant and antifungal activity, confirming that it can be subject to application in pharmaceutical or food industries.

## Figures and Tables

**Figure 1 molecules-27-02949-f001:**
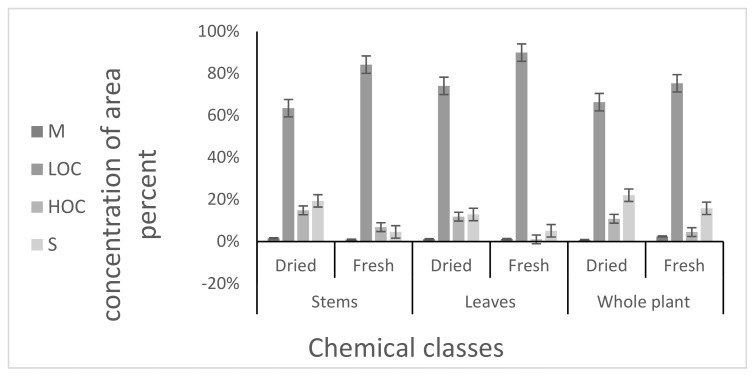
Drying effect on volatile oils constituents of *M. suaveolens* L. Sesquiterpenes (S), heavy oxygenated compounds (HOC), light oxygenated compounds (LOC), and monoterpenes (M).

**Figure 2 molecules-27-02949-f002:**
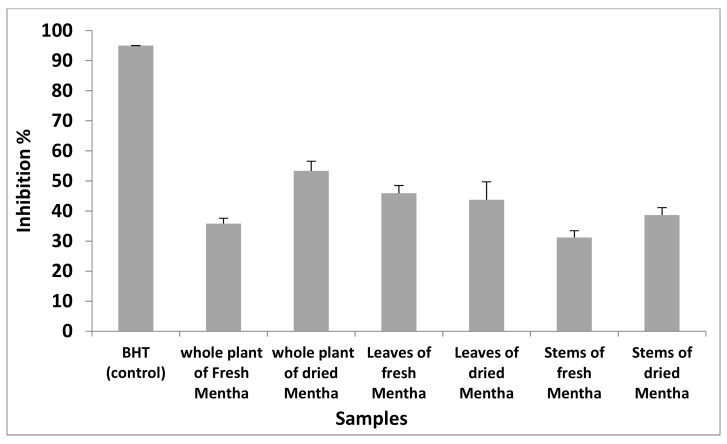
Antioxidant activity of volatile oils of *M. suaveolens* L.

**Figure 3 molecules-27-02949-f003:**
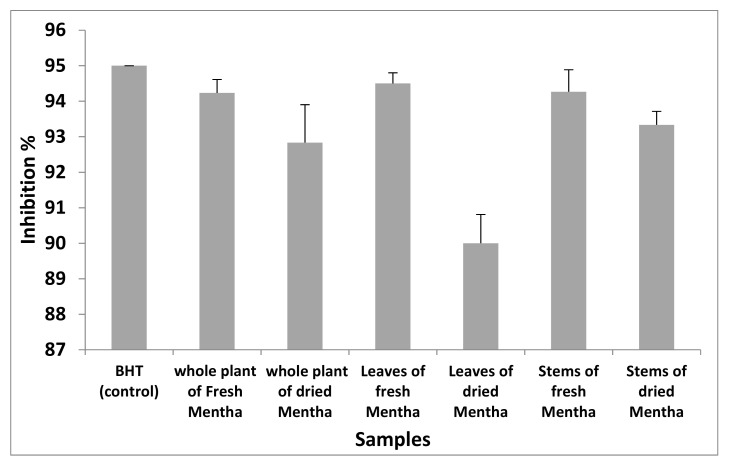
Antioxidant activity of non-volatile constituents of *M. suaveolens* L.

**Figure 4 molecules-27-02949-f004:**
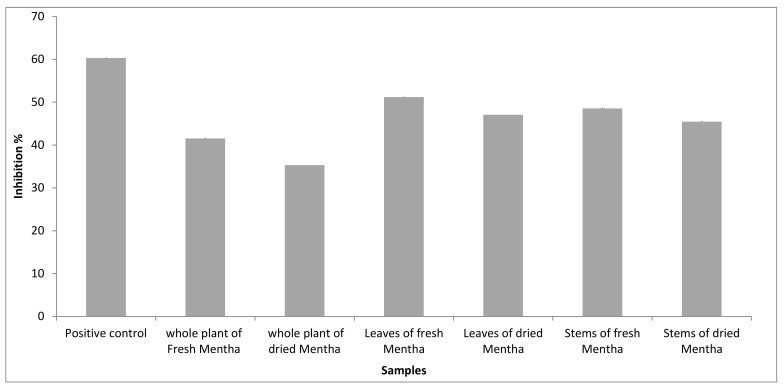
Antifungal activity of volatile oils of *M. suaveolens* L.

**Figure 5 molecules-27-02949-f005:**
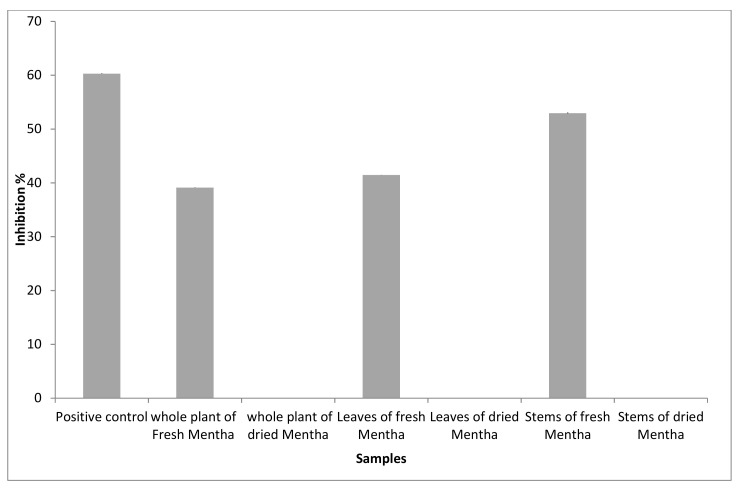
Antifungal activity of non-volatile constituents of *M. suaveolens* L.

**Table 1 molecules-27-02949-t001:** Chemical components of volatile oils from fresh and dried *Mentha suaveolens* L. analyzed by GC/FID and GC/MS.

^a^ Conc. %						^b^ Calculated KI	^c^ KI Data Average	Name	^d^ Type	Identification Methods
Whole Plant		Leaves		Stems						
Fresh	Dried	Fresh	Dried	Fresh	Dried					
0.01	3.22	0.03	0.18	0.01	0.08	978	977	Octen-2-ol	LOC	KI and MS
0.03	0.12	0.02	0.1	0.02	0	989	989	Decan<1>	M	KI and MS and St.
0.07	0.31	0.04	0.14	0.67	0.23	998	999	Ethyl hexanoate	LOC	KI and MS
0.04	0.05	0.02	0	0.34	0.22	1005	1007	Hexenyl acetate(3Ƶ)	LOC	KI and MS
0.07	0.06	0	0.02	0.03	0.02	1007	1007	Hexenoic acid (2E)	LOC	KI and MS
0.09	0.03	0.03	0	0.03	0.01	1008	1008	Linalool oxide (dehydroxy-cis)	LOC	KI and MS
0.1	0.06	0	0.04	0.01	0.01	1011	10,011	δ-Carene (3)	M	KI and MS
0.03	0.03	0	0.01	0.03	0.01	1013	1013	Hexenyl acetate (2E)	LOC	KI and MS
0.54	0.01	0.25	0	0.35	0	1014	1013	1,8 Cineole	LOC	KI and MS and St.
0.39	0.03	0.16	0.02	0.24	0.86	1017	1017	Terpinene(α)	M	KI and MS
0.01	0	0	0	0.01	0.04	1024	1025	*p*-Cymene	M	KI and MS
0.12	0	0	0.08	0	0.08	1026	1026	Menthene (1-p)	M	KI and MS
0.04	0.01	0	0.01	0.03	0.09	1029	1029	β-Phellandrene	M	KI and MS
0.04	0.01	0.01	0.01	0.02	0	1037	1038	Ocimene(Ƶ-β)	M	KI and MS
0.62	0.15	0.33	0.31	0.1	0.18	1050	1048	Ocimene(Ε-β)	M	KI and MS and St.
0.09	0.03	0.04	0.01	0.03	0.01	1059	1060	Terpinene(-γ-)	M	KI and MS
0.13	0.03	0.03	0	0	0.1	1066	1065	Octen-1-ol(2Ε)	LOC	KI and MS
1.71	0.29	0.71	0.56	1.26	0.2	1076	1076	Benzyl formate	LOC	KI and MS and St.
0.81	0.25	0.49	0.48	0.34	0.25	1088	1086	Terpinolene	M	KI and MS and St.
1.54	0.46	1.13	0.9	1.18	0.18	1096	1097	Linalool	LOC	KI and MS and St.
0.03	0	0.01	0.01	0	0.05	1101	1102	Hexyl propanoate	LOC	KI and MS
0.05	0.03	0.04	0.01	0.03	0.01	1104	1103	Methyl butyl isovalerate(2)	LOC	KI and MS
0.13	0	0.08	0.05	0.02	0.05	1113	1115	Camphenol(6-)	LOC	KI and MS
0	0.01	0.01	0.01	0.02	0.03	1121	1122	Menth-2-en-1-ol(cis-p)	LOC	KI and MS
5.88	2.81	3.93	4.98	3.85	2.85	1122	1122	Myrcenol	LOC	KI and MS and St.
4.29	1.84	5.61	2.44	4.4	1.04	1133	1132	Terpineol<1->	LOC	KI and MS and St.
0.22	0.05	0.17	0.09	0.1	0.04	1144	1143	Ocimene(neo-allo)	M	KI and MS
0.02	0.01	0.01	0	0.05	0.04	1153	1155	Thujanol(neo-3)	LOC	KI and MS
1.38	0.23	0.83	0.52	0.13	0.09	1159	1159	Isopulegol(iso)	LOC	KI and MS
0	0.01	0.03	0.03	0	0.08	1160	1159	Isoborneol	LOC	KI and MS
0.09	0.01	0.05	0.02	0.07	0.08	1164	1164	Terpineol(cis dihydro)	LOC	KI and MS
0.03	0.06	0.03	0.01	0.01	0.33	1165	1164	Menthol(neo)	LOC	KI and MS
0.3	0.07	0.17	0.11	0.07	0.15	1170	1170	Pinocampheol	LOC	KI and MS
0.26	0.03	0.13	0.06	0.07	0.06	1171	1171	Iso pulegol (neoiso)	LOC	KI and MS
0.16	0.05	0.09	0.04	0	0.09	1174	1174	Linalool oxide (cis)	LOC	KI and MS
0	0.03	0.01	0.03	0.02	0.05	1176	1176	Linalool oxide (Trans)	LOC	KI and MS
0.09	0.01	0.06	0	0.06	0.01	1177	1177	Terpinen-4-ol	LOC	KI and MS and St.
0.18	0.02	0.06	0.02	0.2	0.05	1186	1187	Dillether	LOC	KI and MS
0.15	0.07	0.08	0.04	0.05	0.01	1188	1188	α-Terpineol	LOC	KI and MS and St.
0.07	0.01	0.04	0.01	0.03	0	1189	1189	Verbanol (neoiso)	LOC	KI and MS
0.02	0.02	0.04	0.01	0.04	0.03	1192	1192	Dihydro carveol	LOC	KI and MS
0.05	0.01	0.01	0.01	0.05	0	1193	1194	Dihydro carveol(neo)	LOC	KI and MS
0.03	0.03	0.01	0	0.03	0.02	1196	1196	Decanol(3-)	LOC	KI and MS
0.27	0.15	0.45	0.23	0.11	0.06	1199	1199	γ-Terpineol	LOC	KI and MS and St.
0.91	0.09	0.44	0.15	0.08	0.08	1200	1201	Dihydro carveol (Trans)	LOC	KI and MS
0.15	0.01	0.11	0.03	0.72	0.03	1208	1208	Piperitol	LOC	KI and MS
0.01	0.03	0.02	0.05	0.9	0.69	1214	1213	Pulegol (Trans)	LOC	KI and MS
0.11	1.62	0	2.52	0.48	1.28	1215	1216	Dihydro myrcenol acetate	LOC	KI and MS
1.77	0.49	3.45	0.88	2.12	0.63	1216	1216	Carveol (Trans)	LOC	KI and MS and St.
0.05	0.01	0.1	0.02	0.05	0.01	1219	1220	Cyclo citral	LOC	KI and MS
0.05	0.07	0.27	0.13	0.01	0.11	1227	1227	Prenyl cyclo pentanone	LOC	KI and MS
0	1.84	2.24	2.65	2.5	0.02	1229	1229	Carveol(cis)	LOC	KI and MS and St.
0.54	0	0	0	0.03	1.69	1230	1232	Mentha-1,8-dien-2-ol (cis-p)	LOC	KI and MS
0.11	0	0	0.01	0	0.01	1234	1235	linalool acetate (tetrahydro)	LOC	KI and MS
43.65	45	64.31	53.45	58.8	49	1234	1234	Carvone	LOC	KI and MS and St.
3.8	4	2.1	0	2.2	0.01	1237	1237	Pulegone	LOC	KI and MS and St.
0.03	0.08	0.01	0	0.04	0.02	1242	1242	Verbenyl acetate(Trans)	LOC	KI and MS
0.12	0.07	0.06	0.02	0.2	0.06	1244	1244	Isomenthene(2-ethyl)	S	KI and MS
0	0.03	0.01	0	0.06	0.05	1247	1247	Carvotan aceton	LOC	KI and MS
0.01	0.02	0.15	0.01	0.05	0.01	1253	1254	Myrtanal (cis)	LOC	KI and MS
0.11	0.02	0	0.02	0.05	0.01	1254	1253	Piperitone epoxide (cis)	LOC	KI and MS
0.09	0.01	0	0.01	0.01	0.01	1255	1254	Piperitone epoxide (trans)	LOC	KI and MS
0.01	0.03	0.01	0.02	0.02	0.11	1256	1256	Sabinene hydrate acetate	LOC	KI and MS
0.34	0	0.13	0	0.32	0	1258	1257	Carvenone	LOC	KI and MS
0.11	0.02	0.03	0.01	0.17	1.26	1261	1262	Myrtanol(Trans)	LOC	KI and MS
0.05	0.01	0.02	0.02	0.07	0.04	1263	1263	Carvonoxide(cis)	LOC	KI and MS
0.05	0.06	0.05	0.01	0.06	0.06	1265	1265	Cauaiacol acetate<o>	LOC	KI and MS
0.03	0.01	0.15	0.01	0.05	0.34	1276	1267	Thujanol acetate(neo-3)	LOC	KI and MS
0.42	0	0	0	0.04	0	1277	1275	Isopulegyl acetate	LOC	KI and MS
0.33	0.46	0.19	0.85	0.22	0.04	1282	1282	Verbenyl acetate(cis)	LOC	KI and MS
0.22	0.08	0.07	0.11	0.08	0.01	1283	1282	Thujanol acetate(neo iso-3)	LOC	KI and MS
0	0.02	0.01	0	0.09	0.02	1285	1285	Terpinen-7-al(α)	LOC	KI and MS
0.09	0.34	0.4	0.02	0.27	0.01	1288	1288	Fenchol(2-ethyl-endo)	LOC	KI and MS
2.79	1.23	1.24	1.85	0.15	0.7	1289	1288	Limonen-10-ol	LOC	KI and MS
0.04	0.26	0.02	0	0.07	0.06	1290	1291	Thymol	LOC	KI and MS
2.66	2.32	1.09	2.14	0.65	1.2	1328	1329	Silphiperfol-5-ene	S	KI and MS
0.1	0.01	0	0.03	0.05	0.01	1333	1334	*cis*-Carvyl acetate	LOC	KI and MS
0.05	0.08	0.02	0.05	0.04	0.02	1336	1336	Presilphiperfol-7-ene	S	KI and MS
0.06	0.03	0.05	0.08	0.02	0.01	1338	1338	Elemene(δ-)	S	KI and MS and St.
1.69	1.29	0.68	1.31	0.44	0.7	1351	1351	Cubebene(α)	S	KI and MS
0.47	0.11	0	0.15	0.01	0.04	1352	1353	Thymol acetate	LOC	KI and MS
0.52	0.26	0.26	0.55	0.25	0.22	1353	1354	Lengipinene(α)	S	KI and MS
0.06	0	0.02	0.02	0.45	0.01	1354	1354	Ethyl nerolate	LOC	KI and MS
0.21	0.06	0.04	0.06	0.45	0.01	1359	1358	Dihydro carveol acetate	LOC	KI and MS
1.41	0.58	0.56	0.95	0.71	0.59	1371	1372	Cyclo sativene	S	KI and MS
0.17	0.02	0	0.06	0	0.08	1372	1373	*p*-Menthane-1,2,3-triol	LOC	KI and MS
1.37	0.6	0.53	0.8	0.13	0.57	1376	1375	Copaene(α)	S	KI and MS and St.
0.53	0.16	0.18	0.29	0.01	0.09	1380	1380	*cis*-Jasmone	LOC	KI and MS
0.04	0.02	0	0.01	0.02	0.04	1381	1381	Patchoulene(β-)	S	KI and MS
0.19	0.01	0.07	0.07	0.05	0	1382	1383	Daucene	S	KI and MS
0.57	0.21	0.23	0.3	0.06	0.24	1388	1388	Cubebene(β-)	S	KI and MS
0.25	0.1	0.07	0.1	0.08	0.13	1390	1391	Longifolene(iso)	S	KI and MS
0.14	0.06	0.07	0.04	0.09	0.07	1391	1392	Elemene(β-)	S	KI and MS
0.07	0.05	0.02	0.04	0.04	0.02	1392	1393	Sativene	S	KI and MS
0.32	0.17	0.16	0.08	0.27	0.1	1400	1400	Longipinene(β-)	S	KI and MS
0.83	0.04	0.35	0.36	0.54	0.14	1402	1403	Funebrene(α)	S	KI and MS
0.04	0.03	0.07	0.03	0.09	0.04	1408	1408	Caryophyllene(Ƶ)	S	KI and MS
0.19	0.12	0.1	0.05	0.1	0.11	1409	1409	Gurjunene(α)	S	KI and MS
0.07	0.03	0	0.03	0.04	0	1412	1412	β-Caryophyllene	S	KI and MS
0.56	0.16	0.29	0.2	0.13	14	1417	1416	Santalene	S	KI and MS
0	0.03	0.05	0.04	0.03	0.07	1419	1418	Caryophyllene(Ε-)	S	KI and MS
0.08	0.01	0.05	0.08	0.07	0.21	1423	1424	Menth-1-on-9-ol acetate	LOC	KI and MS
0.08	0.01	0.05	0.03	0.03	0.29	1430	1430	Ionone(Ε-α)	LOC	KI and MS
4	11	0.03	3.81	0.01	0.08	1431	1431	Copaene(β)	S	KI and MS
0.02	0.02	0.02	0.07	0.04	0.02	1436	1435	Elemene(-γ)	S	KI and MS
0.09	0.03	0.07	0.05	0.05	0.02	1441	1441	Aromadendrene	S	KI and MS and St.
0.03	0.04	0	0.1	0.03	0.08	1444	1444	α-Humulene	S	KI and MS
0.04	0.05	0	0.05	0.06	0.1	1446	1445	γ-Muurolene	S	KI and MS
0.13	0.04	0.1	0.03	0.01	0.02	1447	1447	Cabreuva(A)	S	KI and MS
0.14	0.07	0.07	0.05	0.18	0.13	1451	1451	Himachalene(α)	S	KI and MS
0.04	0.13	0.04	0.09	0.21	0.2	1454	1454	Neryl propanoate	LOC	KI and MS
0.02	0.05	0.03	0.1	0.07	0.01	1456	1457	Carvyl propanoate(Trans)	HOC	KI and MS
0.19	3.63	0.1	0.12	0.14	0.05	1460	1462	Aromadendrene(allo)	S	KI and MS
0.04	0.93	0.03	1.37	0.11	0.54	1466	1467	Dodecanal	S	KI and MS
0.1	0.43	0.07	0.21	0.07	0.05	1469	1469	Ethyl-(2*Ε*,4*Ƶ*)-decadienoate	HOC	KI and MS
0.02	0.72	0.01	1.08	0.07	0.01	1470	1471	Pinchotene acetate	HOC	KI and MS
0.02	0.8	0.05	0	0.02	0.04	1477	1478	Geranyl propanoate	HOC	KI and MS
0.06	0.06	0.02	2.43	0.09	2.81	1478	1478	Allyl decanoate	HOC	KI and MS
0.05	0.11	0.03	0	0.07	0.1	1480	1481	Cabreuva oxide D	HOC	KI and MS
0.01	0.03	0.02	0.08	0.06	0.06	1482	1482	Menthyl lactate	HOC	KI and MS
1.44	0.32	0.19	0.03	1.88	0.17	1515	1516	Gernyl isobutanoate	HOC	KI and MS
0.03	0.24	0.15	0.17	0.14	0.21	1517	1517	Himachalene(α-dehydro-ar)	HOC	KI and MS
0.13	2.68	0.1	2.23	0.13	1.91	1522	1522	Isobornyl isovalerate	HOC	KI and MS
0.13	2.34	0.18	2.86	0.26	7.06	1524	1524	Isobornyl-2-methyl butanoate	HOC	KI and MS
2.5	3.07	0.2	2.67	4.01	2.44	1640	1640	Epi-α-Muurolol	HOC	KI and MS

^a^ concentration of area %, ^b^ KI (Kovat’s index), ^c^ reported Kovats index, [19,20,21,22,23,24,25,26,27,28]; www.webbook.nist.gov (accessed on 30 March 2022), ^d^ Sesquiterpenes (S), heavy oxygenated compounds (HOC), light oxygenated compounds (LOC), and monoterpenes (M). Standard compound (St). Notes: The analytical replicates were performed twice.

**Table 2 molecules-27-02949-t002:** The chemical composition of fresh and dried *M. suaveolens* L. analyzed by using HPLC.

Phenolic Compounds	Concentration (µg/g) ± SD					
	Whole Plant		LEAVES		Stems	
	Fresh	Dried	Fresh	Dried	Fresh	Dried
**Protocatechuic acid**	5.5 ± 0.3	n.d	45.4 ± 3.21	3.5 ± 0.17	19.96 ± 1.56	3.6 ± 1.23
**p-hydroxybenzoic acid**	n.d	3.95 ± 0.23	526.9 ± 21.87	2.23 ± 0.15	n.d	2.62 ± 0.12
**Catechin**	198.3 ± 1.8	85.03 ± 0.98	462.3 ± 12.45	54.19 ± 3.14	1340.4 ± 13.76	68.1 ± 2.56
**Vanilic acid**	n.d	n.d	12.13 ± 1.12	n.d	n.d	n.d
**Cinnamic acid**	0.33 ± 0.01	10.51 ± 1.02	46.3 ± 3.05	15.35 ± 1.16	1.54 ± 0.13	1.84 ± 0.13
**Naringenin**	2.3 ± 0.15	53.22 ± 3.23	13 ± 1.23	7.2 ± 0.54	371.8 ± 3.89	130.3 ± 10.23
**Eugenol**	n.d	n.d	n.d	n.d	86.6 ± 7.34	n.d
**Caffeic Acid**	n.d	13.84 ± 1.08	1141.5 ± 13.35	1.79 ± 0.09	n.d	3.7 ± 023
**Coumaric acid**	0.05 ± 0.001	n.d	51 ± 3.67	n.d	5.8 ± 0.43	n.d
**Ferulic acid**	226.7 ± 3.8	3.89 ± 0.22	1520 ± 17.34	0.63 ± 0.0054	1.92 ± 0.12	1.95 ± 0.12
**Rutin**	676.7 ± 4.45	252.16 ± 12.92	3383.8 ± 15.45	194.7 ± 12.7	194.6 ± 12.43	41.6 ± 2.34
**Luteolin**	122.3 ± 1.17	78.65 ± 4.34	514.9 ± 12.34	83.7 ± 6.42	41.6 ± 3.52	38.6 ± 2.12
**Quercetin**	22.5 ± 1.08	153.8 ± 7.8	377.3 ± 24.20	156.98 ± 11.65	n.d	58.6 ± 6.23
**Rosmarinic acid**	2223.3 ± 9.8	21,191.9 ± 24.8	28,002.5 ± 32.6	15,165.1 ± 17.15	6558 ± 15.25	8378.4 ± 23.75

n.d: not detected. SD standard deviation. Notes: The HPLC analysis was carried out in duplicate

**Table 3 molecules-27-02949-t003:** Mean squares of analysis of variance for antifungal activity of volatile and non-volatile constituents of *M. suaveolens* L.

Source of Variation	df	Volatile Oils	Ethanolic Non-Volatile Extracts
Between	6	182.952 **	2160.532 **
Within	14	0.025	0.013

** indicates significance at 1% probability level.

## Data Availability

Not applicable.

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
