# Peer review of "Investigation of Chemical Compositions and Biological Activities of Mentha suaveolens L. from Saudi Arabia"

_molecules, 2022, doi:10.3390/molecules27092949_

Round 1

Reviewer 1 Report

An interesting formula and I do enjoy reading this type of research. However, it is not publishable in its present form and suffers from various critical issues as outlined below:

There are many typing errors in the text. Check all the text carefully. 

Title : I would suggest changing the title Investigation of Biological Activities and Chemical Compositions of Saudi Mentha  suaveolens L. to Investigation of Chemical Compositions  and Biological Activities of Mentha suaveolens L. from Saudi Arabia

Line 16: change Mentha in italic style

Line 18, 19:  please replace (GC/MS)& Gas Chromatography(GC), with (GC/MS) and Gas Chromatography Flame ionization detector (GC/FID).

Line 21:  please replace GC/MS & GC, with GC/MS and GC/FID

Line 32: please replace M. suaveolens L with M. suaveolens L.

Line 49: please replace gamma-terpinene with g-terpinene

Line 79: please replace GC/MS & GC, with GC/MS and GC/FID

Results:  Please detail the yield obtained for each part of the plant for both extraction processes

Line 87: All components are listed in table 2……maybe table 1? Please carefully check the numbering of tables and figures

Table caption: please change with Table 1. Chemical components of volatile oils from fresh and dried Mentha suaveolens L. analyzed by GC/FID and GC/MS.

Table 1 and 2: how many analytical replicates were performed? Repeatability and variability data are missing.

Line 95: please replace All constituents are recorded in table 2 with  All constituents are reported in table 1.

Line 317: Please add the paragraph "Chemicals" which contains references for all standards and reagents used.

Line 343: regarding the HPLC method used, there is a lack of data for the pure compounds used for the quantitative analysis (to be included in the Chemicals section) and the data of the concentrations used for the calibration curve and linearity.

In addition how many analytical and distillation replicates were performed? Repeatability and variability data are missing.

For Table 1 see attached file

Author Response

Response to reviewer comments

First of all , authors appreciated and thank reviewer for his valuable comments which impact on the quality and potentiate the manuscript

An interesting formula and I do enjoy reading this type of research. However, it is not publishable in its present form and suffers from various critical issues as outlined below:

Comment There are many typing errors in the text. Check all the text carefully.

Response done

Comments Title : I would suggest changing the title Investigation of Biological Activities and Chemical Compositions of Saudi Mentha  suaveolens L. to Investigation of Chemical Compositions  and Biological Activities of Mentha suaveolens L. from Saudi Arabia.

Response done

Comment; Line 16: change Mentha in italic style

Response done

Comment; Line 18, 19:  please replace (GC/MS)& Gas Chromatography(GC), with (GC/MS) and Gas Chromatography Flame ionization detector (GC/FID).

Response done.

Comment; Line 21:  please replace GC/MS & GC, with GC/MS and GC/FID

Response done

Comment; Line 32: please replace M. suaveolens L with M. suaveolens L.

Response done

Comment; Line 49: please replace gamma-terpinene with g-terpinene

Response  done

Comment; Line 79: please replace GC/MS & GC, with GC/MS and GC/FID

Response done

Comment; Results:  Please detail the yield obtained for each part of the plant for both extraction processes

Response added  

Comment;  Line 87: All components are listed in table 2……maybe table 1? Please carefully check the numbering of tables and figures.

Response the tables were renumbered

Comment; Table caption: please change with Table 1. Chemical components of volatile oils from fresh and dried Mentha suaveolens L. analyzed by GC/FID and GC/MS.

Response caption was changed  

Comment; Table 1 and 2: how many analytical replicates were performed? Repeatability and variability data are missing.

Response added.

Comment Line 95: please replace All constituents are recorded in table 2 with  All constituents are reported in table 1

Response  changed .

Comment; Line 317: Please add the paragraph "Chemicals" which contains references for all standards and reagents used.

Response added

.

comment Line 343: regarding the HPLC method used, there is a lack of data for the pure compounds used for the quantitative analysis (to be included in the Chemicals section) and the data of the concentrations used for the calibration curve and linearity.

Response: All standard (authentic compounds of essential oils and phenolic compounds) and chemical reagents were supplied from sigma Aldrich company and used without further purifications.

Comment: In addition, how many analytical and distillation replicates were performed? Repeatability and variability data are missing

Response Due to similarity of obtained data among of replication for each treatment or the results of variance or standard deviation that approximate to zeroes value. thus, presented error bars on figures doesn't appear.

Reviewer 2 Report

The manuscript presents a major characterisation of Saudi Mentha suaveolens L in terms of volatile and non-volatile fractions. The methodology used for that purpose seems to be appropiate.  

However, the presented manuscript has big improvement areas prior to be published. I really suggest to the authors, please re-write all sections of the manuscript. I attached the pdf document with some comments, but you need to be a bette effort. You have the data, still need to improve the writing. 

Examples:

-Introduction: You must to show why is important to compare different parts of the plant and if it's in a fresh or dried sample. You are missing this point. In addition, please do not forget to explore the biological activities of Mentha suaveolens L and their relation with ther volatile and non-volatile components. 

-Materials and methods: driying methodology is missing. Antioxidant actvitiy better to express in IC50 value (IC50. IC50 represents the concentration at which a substance exerts half of its maximal inhibitory effect. This value is typically used to characterize the effectiveness of an antagonist in inhibiting a specific biological or biochemical process.) In that sense, you could represent all the results in a table, instead of separate graphs, to allow comparisson between all samples.

See the pdf document for other corrections. 

Author Response

Response to reviewer comments

First of all , authors appreciated and thank reviewer for his valuable comments which impact on the quality and potentiate the manuscript

Comment; Improved the title. Needs to be more original and innovative.

Response the title changed into (Investigation of Chemical Compositions and Biological Activiities of Mentha suaveolens L. from Saudi Arabia)

Comment Abstract must be improved. Line 20 - to mention which biological activities the reader will find.

Response; The ethanolic extract of fresh stem showed the highest antifungal activity ( 53% inhibition)0 comparing with miconazole ( 60% inhibition) but the ethanoic extract of dry stem showed no activity. Also, all ethanolic extracts whether for fresh or dry Mentha, have antioxidant activity more than 90% while the antioxidant of whole plant volatile oil is equal to 53.33%

Comment The introduction needs to be rewrite.

Response the introduction was rewritten according to reviewer valuable comment

Comment, Please see the Materials and Methods section to correct this error of concept.

Response corrected

Comment A- Concentration %. Please be more specific, percentage of what? Sum of areas of identified compounds or % of total detected peaks?

Response concentration of area %

Comment or all Figures included in the manuscript: The stilying need to be improved. (Decimals, name of vertical axis, propper leyend, in conclusion a better and clear graph must be desing.)Instead representig the value of each bar, error bars must to be included, as well as statistic significance.

Response all figures improved

Comment Include the meaning for M, LOC, HOC, and S. The Figure needs to be explained by it self for the reader

Response done

Comment Standard deviation is missing. ¿How many analyses did you execute? It must be done at least in duplicate.

Response standard deviations were added and the duplicate experimental were done.

Comment Please, it is more appropiate to express the results in IC50 value. So, when you correct that, please represent the results in a table, for better comparisson whitin different samples.

Response authors respect reviewer valuable comment but authors performed this work as extension to pervious work which expressed in the same way as percentage of inhibition.

Comment Why is this section showed? I think this is not neccesary. Also, the discussion section did not refers to this result at all.

Response removed .

Comment Where is the Table S1?

Response corrected

Comment Please be more specific, in this way is not clear. Do you mean Penicillium sp?

Response fungi species Corrections for Figure 4 and 5:

Comment The resolution of the figure is not appropiate, it is difficult to read. ¿Why the error bars are not represented in the graph? Please, named the vertical axis. That's a basic when using graphs.Instead of represent the indivual value, just show the statistical analysis. It's not clear in the present way.

Response done and also Due to similarity of obtained data among of replication for each treatment or the results of variance or standard deviation that approximate to zeroes value . thus presented error bars on figures doesn't appeared

Comment You need to be sure that abbreviations are explained for the first time above, in other case, you need to explain here.

Response abbreviations were added

Comment How the dried sample was obtained? You need to described the methodoly since dried samples are compared with fresh one. Also you need to include an statement to better understand that the calculations are correct. I mean, that the comparison of results are done in a correct way, if the water lossses during drying were taken into account. It's better to express the results as per fresh matter.

Response the plants were dried under shad at rt in fuming cop

Comment More info about de apparatus is missing (origin, name, model, etc)

Response the Clevenger Apparatus was supplied from Shiva Scientific Glass Private Limited, India.

1L Rotary Evaporator (Model: R-3001, Evaporating Flask: 500ml, 1000ml, Rotation Speed: 10-280rpm, Temperature Range: Room temp+5℃ ~95℃, Evaporating Speed: 23.5ml/min (Water) supplied from GWSI manufacture, China.

Comment You don't need to separate GC and GC/MS. Both analyses are complementary and it an be explained together

Response the separation  was only done in methodology but in other part we merged it .

Comment;  How do you identify and quantify the phenolic compounds? In supp materials there is not explained.

Response All phenolic standard were prepared in methanol within range from 0.5 to 100 µg/ml and this standards were used for quantitative and qualitative identification of phenolic com-pounds in M. suaveolens L extracts  chemical composition depending the standards cali-bration curve and retention time respectively.

Comment ; What is mean DPPH, BHT? You`re missing a lot of basic rules about a manuscript writing, like explain the abbreviations when using the first time. Please correct all over the manuscript.

Response all abbreviations were added where firstly appeared

Reviewer 3 Report

The manuscript entitled “Investigation of Biological Activities and Chemical Compositions of Saudi Mentha suaveolens L.” presents the research topic that may be of value for Molecules readers. However, in my opinion some points should be corrected before its acceptance. My general comments on the study are listed in the paragraphs below:

  • In the Introduction section the authors said that: “According to our knowledge there is no report on the chemical composition of suaveolens L volatile oils and biological activities of M. suaveolens L belonging to Saudi Arabia (Aljouf area).”, and while it is true, there are many articles regarding the composition and activity of M. suaveolens essential oils from different geographical regions (e.g. doi:10.4103/1995-7645.254937; doi:10.3390/molecules20058605; doi:10.1002/cbdv.201700320; doi:10.1016/j.heliyon.2020.e05480; doi:0.1590/0001-3765202120190478; doi:10.3109/13880209.2013.865239; and a few other). Therefore, it should be mentioned in the Introduction, together with the explanation of the novelty aspect of the presented research. Based on previous research, did the authors expect that the origin of the material had a significant effect on composition and activity? Because based on the Introduction, it seems that the composition differ between various species, and not within one species. In this context, I would recommend to change the narrative of the Introduction to demonstrate the novelties of the research, if they are.
  • Table 2: What “St.” mean in the identification method? Is it “standard” or something else?
  • There are no standard deviations in studies of composition and activity (tables, figures, text), they should be added.
  • What were the identification methods in the HPLC analysis? Were there any reference compounds? And how eugenol was detected in the hydrophilic extracts?
  • How the compounds contents were determined in the HPLC analysis (what calibration curves)?
  • What was the concentration of extracts/oils in the DPPH analysis? The mere mention that something works in a certain percentage without specifying the concentration does not say much. It would be best to determine the IC50 values or at last to give the concentration.
  • Line 155: There is a mention about Table 3, but in table 3 there are results of HPLC analysis.
  • Line 161: Where is the Table S1? It is not in the supplementary files?
  • Incorrect selection of literature and reasoning in the Discussion section. In most cases, the authors compare the composition obtained for the studied species with the composition of other mint species (literature e.g. 3,4,7,8,25,26,27,29,30,35,36,37,38,39), although usually they do not even mention which species, using only the term "Mentha".This is misleading and suggests comparing data from the same species. It is also not appropriate, since the data should be compared first for the same species, if there are any (and in this case there are many such publications), and then for other species (but it should be clearly stated which ones). Therefore, the sentence “Based on the above results, significant change in the chemical composition of volatile oils might be due to different plant parts, environmental factors, seasons, geographical and plant age of the Mentha plant. (line 253-255)”, is only partly correct, because I would say that change in the chemical composition is mainly due to comparison with different species. Similarly in case of activity testing – there is the sentence “These results agree with Mata et al. [44], who reported that ethanol extracts have good antioxidant activity. (lines 293-294)”, but for what species? Because not all ethanolic extracts will be active.

Minor points:

  • Lines 25, 27, 42, 70, 72, 75, 102, 109, 404, 405: additional spaces
  • Line 47: wrong word “wither”
  • Line 51: wrong word “their”
  • Lines 68-69: “Also, Copaene(β) 68 was found to have good antioxidant activity [11].” The reference to the activity of this compound is unclear at this stage.If it is a compound previously detected in the volatile oils of suaveolens, it should be clearly mentioned.
  • Line 74: non-volatile fractions
  • Table numeration: The first table is “Table 2”, so where is table 1?
  • Lines 129-130: the part of the subtitle was transfer to the first paragraph
  • Line 130-134: it sounds like the method description, not results.
  • Line 158: what is the https address?

Author Response

Response to reviewer comments

First of all , authors appreciated and thank reviewer for his valuable comments which impact on the quality and potentiate the manuscript

Comment1;  In the Introduction section the authors said that: “According to our knowledge there is no report on the chemical composition of M. suaveolens L volatile oils and biological activities of M. suaveolens L belonging to Saudi Arabia (Aljouf area).”, and while it is true, there are many articles regarding the composition and activity of M. suaveolens essential oils from different geographical regions (e.g. doi:10.4103/1995-7645.254937; doi:10.3390/molecules20058605; doi:10.1002/cbdv.201700320; doi:10.1016/j.heliyon.2020.e05480; doi:0.1590/0001-3765202120190478; doi:10.3109/13880209.2013.865239; and a few other). Therefore, it should be mentioned in the Introduction, together with the explanation of the novelty aspect of the presented research. Based on previous research, did the authors expect that the origin of the material had a significant effect on composition and activity? Because based on the Introduction, it seems that the composition differ between various species, and not within one species. In this context, I would recommend to change the narrative of the Introduction to demonstrate the novelties of the research, if they are.

Response , the introduction was rewritten and  recommended references were added

Comment2,  Table 2: What “St.” mean in the identification method? Is it “standard” or something else?

Response; St means standard compounds and added under  table.

Comment 3; There are no standard deviations in studies of composition and activity (tables, figures, text), they should be added.

Response added

Comment 4; What were the identification methods in the HPLC analysis? Were there any reference compounds? And how eugenol was detected in the hydrophilic extracts?

Response methods were added also the eugenol is phenolic compound so can be detected in hydrophilic extract.

Comment 5;  How the compounds contents were determined in the HPLC analysis (what calibration curves)?

Response; All phenolic standard were prepared in methanol within range from 0.5 to 100 µg/ml and this standards were used for quantitative and qualitative identification of phenolic com-pounds in M. suaveolens L extracts  chemical composition depending the standards cali-bration curve for each standard and retention time respectively.

Comment 6;  What was the concentration of extracts/oils in the DPPH analysis? The mere mention that something works in a certain percentage without specifying the concentration does not say much. It would be best to determine the IC50 values or at last to give the concentration.

Response; DPPH solution (0.1 mM, 1.8 mL) was mixed with 0.2 mL of each diluted Mentha extract stock solution (5 mg/mL) and incubated in the dark for 30 min at 25 °C, (  added to material and methods, antioxidant activity ).

Comment 7; Line 155: There is a mention about Table 3, but in table 3 there are results of HPLC analysis.

Response the numbering of tables is corrected.

Comment 8; Line 161: Where is the Table S1? It is not in the supplementary files?

Response corrected

comment 9;  Incorrect selection of literature and reasoning in the Discussion section. In most cases, the authors compare the composition obtained for the studied species with the composition of other mint species (literature e.g. 3,4,7,8,25,26,27,29,30,35,36,37,38,39), although usually they do not even mention which species, using only the term "Mentha". This is misleading and suggests comparing data from the same species. It is also not appropriate, since the data should be compared first for the same species, if there are any (and in this case there are many such publications), and then for other species (but it should be clearly stated which ones). Therefore, the sentence “Based on the above results, significant change in the chemical composition of volatile oils might be due to different plant parts, environmental factors, seasons, geographical and plant age of the Mentha plant. (line 253-255)”, is only partly correct, because I would say that change in the chemical composition is mainly due to comparison with different species.

Similarly in case of activity testing – there is the sentence “These results agree with Mata et al. [44], who reported that ethanol extracts have good antioxidant activity. (lines 293-294)”, but for what species? Because not all ethanolic extracts will be active.

Response Egyptian and Moroccan essential oil of M. suaveolens L. were studied and the results shown that β-copaene is present in few amount (less than 1%) [Ref]. Also, copaene had been detected in some Algerian species of Mentha with concentration ranging from 0.82 to 0.9% [Ref]. Boukhebti et al. shown that the content of copaene in the essential oil of Mentha spicita was 0.347% . Other study conducted in USA shown that beta copaene was present in Peppermint EO (0.07%), Native Spearmint (0.25%), Scotch Spearmint (0.18%).

In this study, α and β copaene had been detected in whole plant, leaves and stems where the concentration of α copaene (0.57%, 0.8%, and 1.37% respectively) and β copaene (11%, 3.81% and 0.08% respectively).

Minor comments

comment 1 Lines 25, 27, 42, 70, 72, 75, 102, 109, 404, 405: additional spaces

response corrected

comment 2 Line 51: wrong word “their”

response corrected .

comment 3; Lines 68-69: “Also, Copaene(β) 68 was found to have good antioxidant activity [11].” The reference to the activity of this compound is unclear at this stage. If it is a compound previously detected in the volatile oils of M. suaveolens, it should be clearly mentioned.

Response. The reference for antioxidant activity of copaene is added.

comment 4; Line 74: non-volatile fractions

response corrected to non-volatile extracts

comment 5• Table numeration: The first table is “Table 2”, so where is table 1?

Response Corrected

Comment 6 • Lines 129-130: the part of the subtitle was transfer to the first paragraph.

Response corrected

Comment 7 • Line 130-134: it sounds like the method description, not results.

Response; corrected

  • comment 8; Line 158: what is the https address?

Response; It is the molecular identification (Modern identification of microorganism)or  fungus that used in the study

Reviewer 4 Report

Manuscript molecules-1685651   Investigation of Biological Activities and Chemical Compositions of Saudi Mentha suaveolens L.  by Bashayr Aldogman , Hallouma Bilel , Shaima Mohamed Nabil Moustafa , Khaled Farouk El-Massry , Hazim M Ali , Faddaa Qayid Alotaibi , Mohamed Hamza , Mohamed A Abdelgawad , Ahmed H. El-Ghorab * is devoted to investigation of chemical composition and some kinds of activity samples from Saudi Mentha suaveolens L.  chemical composition of essential oil is well investigated by many authors but such investigations for non volatile compounds from the whole plant, stems and leaves are rare in scientific literature. Methods used are correct and in accordance with modern trends of phytochemical investigation. Article is enough well organised. I have some advise for discussion. 1. All chemicals and standards used should be listed in the section Materials and methods. Method of extracts preparation for biological investigation should be described too. 2. Manuscript and supplementary materials don't contain data about methods validation 3. I reccommend add to introduction more information about chemical composition of Mentha suaveolens L. from Spain, Algeria, Lithuania etc described before  4. I think discussion need not only information about chemical composition but some comparison - between different Mentha species, between samples from different countries, between different parts of plant. It will be interestiong for choosing in further plant raw material for pharmacological investigation and may be pharmaceutical development. The same about biological activity. Discussion should be corrected and completed 5. In my opinion its not correct in this experiment to propose which components influence to activities. It may be stated only after biological investigation of individual compounds in the same experimental conditions. For antioxidant one there is HPLC method with postcolumn derivatisation (

Marksa, M.Radušiene, J.Jakštas, V.Ivanauskas, L.Marksiene, R. Development of an HPLC post-column antioxidant assay for Solidago canadensis radical scavengers , Natural Product Research2016, 30(5), стр. 536–543)

6. I don't know if its correct to conclude about preservative perspectives of samples because cheese fungi used are very specific.  7. Conclusions should be rewritten with noting about novelty, practical and scientific soundness of investigations done.

Author Response

Response to reviewer comments

First of all , authors appreciated and thank reviewer for his valuable comments which impact on the quality and potentiate the manuscript

Manuscript molecules-1685651   Investigation of Biological Activities and Chemical Compositions of Saudi Mentha suaveolens L.  by Bashayr Aldogman , Hallouma Bilel , Shaima Mohamed Nabil Moustafa , Khaled Farouk El-Massry , Hazim M Ali , Faddaa Qayid Alotaibi , Mohamed Hamza , Mohamed A Abdelgawad , Ahmed H. El-Ghorab * is devoted to investigation of chemical composition and some kinds of activity samples from Saudi Mentha suaveolens L.  chemical composition of essential oil is well investigated by many authors but such investigations for non volatile compounds from the whole plant, stems and leaves are rare in scientific literature. Methods used are correct and in accordance with modern trends of phytochemical investigation. Article is enough well organised. I have some advise for discussion.

  1. All chemicals and standards used should be listed in the section Materials and methods. Method of extracts preparation for biological investigation should be described too.

Response; all chemicals and standards   (authentic compounds of essential oils [Decan, 1,8 Cineole, Oci-mene(Ε-β), Benzyl formate, Terpinolene, Linalool, Myrceno, Terpineol<1->, Terpinen-4-ol, α-Terpineol, γ-Terpineol , Carveol (Trans Carveol(cis), Carvone, Pulegone, Elemene(δ-), Copaene(α), Aromadendrene] and phenolic compounds [Protocatechuic acid, p-hydroxybenzoic acid, Catechin, Vanilic acid, Cinnamic acid, Naringenin, Eugenol, Caf-feic Acid, Coumaric acid, Ferulic acid, Rutin, Luteolin, Quercetin and Rosmarinic acid]) were added to material and methods; and also the description of extract preparation and concentration for biological assessment  were added (for antifungial, One ml of plant extract (20 mg/ml) was added to each well to determine the inhibition zone and inhibition percentage (%). But for antioxidant, the scavenging DPPH free radical ability was determined in vitro according to the procedure of Yue and Xu [59]. DPPH solution (0.1 mM, 1.8 mL) was mixed with 0.2 mL of each diluted Mentha extract stock solution (5 mg/mL) and incubated in the dark for 30 min at 25 °C In this assay, the percentage of DPPH reduction by samples was compared to BHT. The experiment was repeated three times. The following formula was used to quantify radical scavenging activity):

  1. Manuscript and supplementary materials don't contain data about methods validation.

Response the authors followed adjusted  reported method[ reference]

  1. I reccommend add to introduction more information about chemical composition of Mentha suaveolens L. from Spain, Algeria, Lithuania etc described before.

Response added

  1. I think discussion need not only information about chemical composition but some comparison - between different Mentha species, between samples from different countries, between different parts of plant. It will be interestiong for choosing in further plant raw material for pharmacological investigation and may be pharmaceutical development. The same about biological activity. Discussion should be corrected and completed.

Response added

  1. In my opinion its not correct in this experiment to propose which components influence to activities. It may be stated only after biological investigation of individual compounds in the same experimental conditions. For antioxidant one there is HPLC method with postcolumn derivatisation . Marksa, M., Radušiene, J., Jakštas, V., Ivanauskas, L., Marksiene, R. Development of an HPLC post-column antioxidant assay for Solidago canadensis radical scavengers , Natural Product Research, 2016, 30(5), стр. 536–543.

Response, authors agree with valuable reviewer comment and  authors attributed the biological activity of each extract to literature survey about the biological activity of each compounds also author prefer to deal with extract in biological activity regarding to natural samples

  1. I don't know if its correct to conclude about preservative perspectives of samples because cheese fungi used are very specific.

Response The fungi that isolated from cheese, can be also isolated from different sources like soil, water, juice and so on.

In this study, we recommended to use this substance as preservative natural substance against different microbes not only  the isolated fungus in this study, as well we recommended several study to make sure that the effect of this substance extended to anther microbes

Finally this substance are natural that have not any side effect to human

  1. Conclusions should be rewritten with noting about novelty, practical and scientific soundness of investigations done

Response; conclusion was rewritten

Round 2

Reviewer 2 Report

Dear authors.

I'm glad to see that recommendations were accepted to improved your publication. Still, the design of the graphs are not correct to me, they lack on professionalism and academic style. Please correct the style:

-do not need to use background color (Fig. 2,3 and 4). Also, not use pattern just gray or black/white scale, be careful with the type of font that you are using

-do not need decimals in the numbers in fig. 1, also erase the secondary lines in the background and add the principal line in Y axis.

- For all Figures, erase the presented data over each bar, ).

Look other similar publications as example. 

You can improved these little details to have a more professional presentation of the results.

Author Response

Response to reviewer 2 comments

First of all, authors thank reviewer for his respected comments which affect on the property and potentiate the manuscript

I'm glad to see that recommendations were accepted to improve your publication. Still, the design of the graphs is not correct to me, they lack on professionalism and academic style. Please correct the style:

  • Comment 1; Do not need to use background color (Fig. 2,3 and 4). Also, not use pattern just gray or black/white scale, be careful with the type of font that you are using

Response done

  • Comment 2; Do not need decimals in the numbers in fig. 1, also erase the secondary lines in the background and add the principal line in Y axis.

Response done

  • Comment 3; For all Figures, erase the presented data over each bar, ).Look other similar publications as example.

Response done

Reviewer 3 Report

I believe the manuscript has been sufficiently improved.

Author Response

thanks for your effort and valuable comments which improve the article quality 

Reviewer 4 Report

In general, the authors took into account the comments of the reviewers and made appropriate changes to the article.

I think, supplementary material file should contain data (tables, chromatograms) of method validation

Author Response

First of all, authors thank reviewer for his respected comments which affect on the property and potentiate the manuscript quality

In general, the authors took into account the comments of the reviewers and made appropriate changes to the:

  • Comment 1; I think, supplementary material file should contain data (tables, chromatograms) of method validation

Response added to supplementary
